# Telomere Dynamics in Post-Traumatic Stress Disorder: A Critical Synthesis

**DOI:** 10.3390/biomedicines13020507

**Published:** 2025-02-18

**Authors:** Ravi Philip Rajkumar

**Affiliations:** Department of Psychiatry, Jawaharlal Institute of Postgraduate Medical Education and Research (JIPMER), Puducherry 605006, India; jd0422@jipmer.ac.in; Tel.: +91-413-229-6280

**Keywords:** stress disorders, post-traumatic, telomere, telomerase, psychological trauma, cellular senescence, biomarkers

## Abstract

Post-traumatic stress disorder (PTSD), a mental disorder caused by exposure to traumatic stress, affects 5–10% of the world’s population. There is some evidence that PTSD is associated with accelerated cellular aging, leading to an increased risk of medical and neurodegenerative comorbidities. Alterations in telomere length (TL) and telomerase enzyme activity have been proposed as biomarkers of this process. This hypothesis was seemingly confirmed in preliminary research, but more recent studies have yielded mixed results. The current narrative review was conducted to provide a critical synthesis of existing research on telomere length and telomerase in PTSD. Data from 26 clinical studies suggest that TL in PTSD is highly variable and may be influenced by methodological, demographic, trauma-related, and psychosocial factors. There is no evidence for altered telomerase activity in PTSD. In contrast, animal research suggests that exposure to traumatic stress does lead to TL shortening. Overall, it is likely that TL is not, by itself, a reliable biomarker of cellular aging in PTSD. Other markers of cellular senescence, such as epigenetic changes, may prove to be more specific in measuring this process in patients with PTSD.

## 1. Introduction

Post-traumatic stress disorder (PTSD) is a mental disorder affecting approximately 5–10% of the world’s population. PTSD develops following exposure to severe or life-threatening traumatic events, such as physical or sexual assault, combat experiences, accidents, or disasters. The symptoms of PTSD include repeated and intrusive recollections of the traumatic incident, increased vigilance and arousal, emotional “numbing” or withdrawal, and avoidance of situations that could trigger memories of the incident [1]. Historically, PTSD was understood as a psychological response to severe trauma, particularly in the context of armed combat. For this reason, it was formerly known as “shell-shock”, “combat fatigue”, or “war neurosis” [2]. The contemporary understanding of PTSD is that it is a psychophysiological disorder that can affect both military personnel and civilians. PTSD does not inevitably follow any exposure to trauma. The likelihood of developing this disorder is influenced by three sets of factors: (a) innate vulnerabilities, which may be either genetic or related to early life adversity, (b) the duration and severity of trauma exposure, and (c) factors associated with or following the trauma, such as physical injury, poor social support, stigmatization, or a lack of access to mental health care [3]. PTSD tends to run a chronic course. About 45–55% of patients remain symptomatic when followed up for three to five years, and many of these patients experience significant disability and functional impairment [4,5]. PTSD is a significant contributor to the global burden of disease in populations exposed to war or civil unrest [6].

PTSD is usually treated with a combination of pharmacotherapy and psychotherapy. First-line pharmacotherapy consists of selective serotonin reuptake inhibitors (SSRIs) such as sertraline and paroxetine. Second-line pharmacological agents include the alpha_1_-adrenergic receptor antagonist prazosin, atypical antipsychotics, and other antidepressants such as venlafaxine. Psychological interventions include trauma-focused therapies such as cognitive-behavioral therapy (CBT) and eye movement desensitization and reprocessing (EMDR) [7,8]. About 40% of patients with PTSD experience persistent symptoms despite access to the best available treatments [5]. The past three decades have seen an exponential growth of research into the neurobiological and psychological mechanisms underlying this disorder in the hope that this would lead to safer and more effective therapeutic strategies [9,10,11].

### 1.1. Pathophysiology of Post-Traumatic Stress Disorder

The neural and systemic alterations associated with traumatic stress and PTSD are complex [10,11,12]. Patients with PTSD exhibit alterations in specific brain regions, circuits, and neurotransmitter activity, as well as more widespread physiological changes [6,10]. For this reason, PTSD is sometimes considered a “systemic” rather than a purely “psychiatric” disorder [13]. Adding to this complexity, some individuals develop PTSD following a single traumatic exposure (e.g., a natural disaster or assault), while others exhibit symptoms in the context of an ongoing, chronic exposure to trauma (e.g., intimate partner violence, repeated exposure to combat). The latter group of patients may differ from the former in terms of their neuroendocrine profile, though they overlap in terms of neural circuit changes [14].

PTSD is characterized by altered transmission in key cortical, subcortical, and brainstem regions and the circuits connecting them. Key brain regions implicated, based on structural and functional brain imaging research, include the ventromedial and dorsolateral prefrontal cortex, anterior cingulate cortex, hippocampus, amygdala, nucleus accumbens, thalamus, and the ventral tegmental area and periaqueductal gray of the midbrain [15,16,17,18,19,20]. Magnetic resonance spectroscopy (MRS) studies of patients with PTSD have found lower levels of N-acetylaspartate (NAA), a marker of neuronal functioning, in the hippocampus and anterior cingulate [21]. Dysfunction of these circuits is linked to alterations in specific neurotransmitters, including glutamate, gamma-aminobutyric acid (GABA), dopamine, serotonin, and neuropeptides [10,19,21]. Distinct neural circuits may be involved in the pathogenesis of the different symptoms of PTSD. Symptoms of emotional numbing may reflect altered functioning in reward processing [20]. Dissociative symptoms and exaggerated threat perception may be related to alterations in sensorimotor processing regions that map the individual’s peri-personal space [22]. On the other hand, fear-related symptoms have been correlated with altered signaling in the “fear network” connecting prefrontal cortical regions with the hippocampus and amygdala [23].

Systemic physiological alterations in PTSD include changes in immune-inflammatory and endocrine activity. Patients with PTSD exhibit an imbalance between pro- and anti-inflammatory cytokines, characterized by elevated peripheral levels of interleukin-1β (IL-1β), interleukin-6 (IL-6), interferon-gamma (IFN-γ), tumor necrosis factor-alpha (TNF-α), and C-reactive protein (CRP). These changes are associated with neuroinflammation and altered functioning of the brain circuits mentioned above [24]. Dysfunction of the hypothalamic-pituitary-adrenal (HPA) axis, involved in the transduction of stress responses, is a key link between neural and systemic inflammatory changes in PTSD. Patients with PTSD exhibit lower basal levels of cortisol, increased glucocorticoid receptor responsiveness, and impaired cortisol responses to experimental stressors. These changes alter not just immune-inflammatory pathways but levels of centrally active neurosteroids and other hormones [25,26]. These systemic changes are associated with the emergence of autoimmune disorders and other medical comorbidities in patients with PTSD [27].

PTSD, like most mental disorders, is best explained by a stress-diathesis model in which innate genetic vulnerability interacts with stressors and other environmental factors. This leads to epigenetic changes in the expression of specific key genes, leading to alterations in specific neural and physiological pathways that are associated with the symptoms of PTSD [28]. Initially, such interactions were studied at the level of single genes. For example, a specific functional polymorphism of the *FKBP5* gene, which regulates glucocorticoid receptor sensitivity, interacts with early life adversity to increase susceptibility to PTSD [29]. More recently, genome-wide analyses have identified 95 genetic variants that confer vulnerability to PTSD, involving 43 specific genes with diverse neural or systemic functions [30]. Patients with symptoms of PTSD show altered expression of some of these genes, reflecting trauma-induced epigenetic changes [31].

### 1.2. Progression, Accelerated Aging, and Neurodegeneration in PTSD

An active area of research and controversy in PTSD is whether the neurobiological changes seen in this disorder only reflect innate vulnerabilities, as in the stress-diathesis model, or are the consequence of ongoing PTSD symptoms. The latter view, known as the neurotoxic stress theory (NST), posits that the recurrent experience of PTSD symptoms, or repeated exposure to similar forms of trauma, causes progressive neuroanatomical and neurophysiological changes [32]. Evidence in support of the NST is mixed. There does appear to be a subset of patients with PTSD characterized by the following changes: (a) persistence or worsening of symptoms, (b) progressive structural or functional changes in frontal lobe subregions, (c) cognitive impairment, (d) poorer functional outcomes, and (e) worse physical health. This has sometimes been referred to as the “neuroprogression” of PTSD [33]. Adverse physical health outcomes associated with PTSD include autoimmune, musculoskeletal, gastrointestinal, endocrine, and cardiovascular diseases: thus, it may be more accurate to speak of “systemic progression” or simply “progression” rather than “neuroprogression”. The pathogenesis of these physical complications probably involves alterations in neuroendocrine and immune-inflammatory pathways, leading to increased oxidative stress and tissue damage [34,35,36,37].

Many of the medical comorbidities seen in PTSD, such as systemic hypertension, ischemic heart disease, type II diabetes mellitus, and peptic ulcer disease, are similar to those associated with aging [37,38]. Over the past two decades, evidence has accumulated that PTSD may be associated with accelerated aging at a cellular level. In 2011, Miller and Sadeh hypothesized that the “re-experiencing” symptoms of PTSD lead to recurrent activation of brain circuitry related to fear responses, increased peripheral activity of the sympathetic nervous system, and dysregulation of the hypothalamic-pituitary-adrenal axis. This is hypothesized to lead to increased oxidative stress, which is associated with cellular aging through its effects on telomere biology at the cellular level [39].

Another strand of evidence that supports the “accelerated aging” hypothesis is the association between PTSD and neurodegenerative disorders, particularly dementia. Patients with PTSD have cognitive impairments across multiple domains, particularly affecting attention and memory. These deficits are associated with alterations in frontal, temporal, and parietal cortical volumes, impaired cerebral white matter integrity and functional connectivity, and reduced levels of brain-derived neurotrophic factor (BDNF) [40,41]. In some cases, these deficits progress to dementia. A recent meta-analysis found that PTSD was associated with a 1.5–1.6-fold increase in the risk of all-cause dementia. This association was almost twice as strong in civilians as in military veterans, suggesting that it was independent of combat-related traumatic brain injuries [42]. A number of cellular mechanisms have been proposed to account for the increased risk of dementia in PTSD; it has been suggested that the final common pathway through which these mechanisms act is accelerated cellular aging [39].

### 1.3. Telomere Biology in Relation to Accelerated Aging in PTSD

#### 1.3.1. Telomere Length and Telomerase in Relation to Aging and Stress

The final common pathway through which stress is thought to influence aging is alterations in telomere biology [43]. Telomeres are tandem repeating sequences of DNA, with the nucleotide sequence TTAGGG, that coexist with a protein complex at the ends of chromosomes. Shortened telomere length is a reliable marker of cellular aging and senescence. Telomerase is a ribonucleoprotein complex composed of the telomerase reverse transcriptase (TERT) protein, a telomerase RNA component (TERC), and several associated proteins. The function of telomerase is to catalyze telomere synthesis and lengthening, leading to increased cellular longevity and protection from aging and apoptosis [44,45,46].

A wide range of factors can influence telomere length (TL). TL is moderately to highly heritable in vertebrates (estimated heritability 25–65%) and humans (estimated heritability 34–82%) [47,48,49]. Whole-genome analysis has identified at least 64 variants in thirty genes that influence TL [50]. Apart from genetics, TL is influenced by a wide range of environmental factors. These factors influence TL through epigenetic changes, involving the methylation of multiple specific genes, whose pattern may vary according to gender and ethnicity [51]. The environmental factors that can reduce TL include diet, physical activity, early life adversity, ongoing stress, and exposure to environmental toxins or drugs of abuse [52,53,54,55,56,57,58,59,60,61]. Maternal exposure to stress during pregnancy, but not earlier, has been associated with reduced TL in offspring [62]. On the other hand, certain healthy lifestyle practices, such as a Mediterranean dietary pattern or meditation, have been shown to protect against telomere erosion [52,63]. Due to the wide range of factors affecting the single “outcome” of TL, careful multivariate analyses and correction for confounding factors are required when estimating the effect of any single variable on TL [64,65,66,67,68]. Variations in study methodology, including the specific assay used and the tissue being studied, can also affect measurements of TL in clinical research [69,70,71].

For all these reasons, TL is only weakly correlated with measures of aging such as physical strength and cognitive test performance. The correlation between TL and chronological age is also modest (*r* = −0.3). Thus, though TL is significantly associated with age, mortality, and age-related illnesses, it cannot be considered a “predictive biomarker of aging” [71].

As the principal regulator of TL, telomerase enzyme activity is affected by many of the same factors that affect TL. These include age, genetic variants, dietary constituents, and physical exercise [72,73,74,75]. Telomerase activity is also significantly influenced by stress. Brief stressors cause a modest increase in telomerase activity, whereas chronic stress is associated with reduced activity [76,77]. The latter finding probably accounts for the link between chronic stress and reduced TL [78]. It should also be noted that there are telomerase-independent processes that can alter TL, known as the alternative lengthening of telomeres (ALT) mechanism [79]. However, the relationship between psychosocial stress and ALT is unknown.

#### 1.3.2. Telomere Length and Telomerase Activity in Mental Disorders

Despite the aforementioned limitations, TL has been extensively studied as a putative marker of cellular aging in patients with mental disorders. There is also a smaller body of research examining telomerase activity in these disorders. Two main factors have driven research on telomere biology in relation to mental illness. First, many mental disorders arise in relation to stress, which is associated with telomere shortening or erosion [58,78]. Second, a specific relationship between certain mental disorders, such as depression and schizophrenia, and accelerated aging has been hypothesized by some researchers and has been tentatively confirmed using other measures of aging [80].

The available evidence on telomere length in relation to mental disorders was evaluated in two systematic reviews. The first of these, conducted in 2016, included data from 27 studies involving 14,827 patients. A significant reduction in leukocyte telomere length (LTL) was observed in a wide range of mental disorders, including mood disorders, schizophrenia, anxiety disorders, and PTSD. The overall magnitude of this effect was moderate (Hedges’ *g* = −0.5) when a meta-analysis was conducted [81]. The second, conducted in 2021, included data from 56 studies involving 113,699 patients but did not include a meta-analytic component. This review concluded that there was evidence of telomere shortening in patients with depression and anxiety disorders, whereas mixed results were observed in patients with PTSD [82]. A large biobank-based study, involving data on 308,725 subjects, also found evidence that persons with depression and, possibly, anxiety disorders had shorter telomeres than those without a mental disorder [83]. A meta-analysis of patients with schizophrenia and related psychoses found small to medium reductions in TL that were independent of age or duration of psychosis [84]. These results provide partial support for the hypothesis of an inverse relationship between TL and mental disorders. This relationship appears to be stronger and more consistent for stress-related conditions.

Apart from these studies, researchers have also attempted to examine the relationship between TL or telomerase and illness-related variables, such as specific symptoms or response to pharmacological treatment in patients with specific psychiatric diagnoses. However, such research is still in its early stages [85,86,87,88,89,90].

#### 1.3.3. Telomere Length and Telomerase Activity in Patients with PTSD

As noted above (Section 1.2), Miller and Sadeh hypothesized that PTSD was associated with accelerated aging. According to their model, the final step in this process is the inhibition of telomerase activity caused by chronic inflammation and oxidative stress, which reduce *TERT* gene expression [39,91]. Oxidative stress can also cause direct damage to telomeres [92]. Lohr et al. reviewed the existing research on TL, inflammatory markers, oxidative stress, and medical comorbidity profiles in patients with PTSD. They also concluded that PTSD was associated with “premature senescence”. However, their interpretation of the results was more cautious: they stated that the exact mechanisms of telomere erosion in PTSD were unknown [38]. A recent meta-analysis of inflammatory and oxidative stress markers in PTSD justifies this caution: while the inflammatory markers CRP, IL-6, and TNF-α were significantly increased in patients with this disorder, no specific oxidative stress marker differed between patients and controls [93]. Thus, an alternative possibility would be that telomere shortening is the result of chronic low-grade inflammation, independent of oxidative stress. There is some in vivo support for this hypothesis. Stressful life events have been associated with concurrent increases in IL-6 signaling and reduced TL [94], and TNF-α levels were monotonically associated with TL shortening both in persons with bipolar disorder and in healthy controls [95].

Telomere shortening in patients with PTSD, whether due to inhibition of telomerase or to other mechanisms, may occur in relation to (a) the primary diagnosis itself, (b) other comorbid mental disorders, such as depression, (c) substance misuse as a form of “self-medication” for severe PTSD symptoms, or (d) the psychosocial consequences of PTSD, such as unemployment or interpersonal relationship problems. More than one of these sets of factors can be operative in a given patient, making it difficult to establish a specific association between PTSD and TL [59,60,81,82,96].

Three systematic reviews, two of which included a meta-analytic component, have examined the evidence for shortened TL in patients with PTSD. However, two of these, discussed above, examined PTSD only as a secondary objective. In the first, which included five studies of patients with PTSD, it was found that LTL shortening was more prominent in patients with PTSD (Hedges’ *g* = −1.3) than in those with other disorders (Hedges’ *g* = −0.2 to −0.6) [81]. In the second, which included only patients with a diagnosis of PTSD, five studies involving 3851 subjects were evaluated. These five studies were the same as those included in the earlier meta-analysis. In this analysis, PTSD was associated with a standardized mean decrease of −0.19 in telomere length. There was no significant gender difference in telomere shortening. Among trauma types, sexual assault and childhood abuse were more specifically linked to reduced telomere length. The authors’ interpretation of these results was in line with the Miller-Sadeh and Lohr et al. reviews: they suggested that oxidative stress, chronic inflammation, and medical or psychiatric comorbidities could all contribute to the telomere shortening seen in this disorder [97]. The third systematic review, which did not include a qualitative synthesis or meta-analysis, examined thirteen studies of 5237 patients with PTSD. This review found more equivocal results: PTSD was associated with telomere shortening in only six of the included studies, and there were significant confounding effects related to early childhood adversity, self-reported hostility, and overall severity of non-specific psychological distress. The authors of this review also noted that studies of military personnel were more likely to find no association between TL and PTSD [82].

In summary, though two earlier meta-analyses appeared to confirm a link between PTSD and shortened TL, more recent literature suggests that the precise links between PTSD and TL are still unclear. Caution is needed in appraising the two “positive” meta-analyses due to certain methodological concerns. These include a relatively small number of studies focused on PTSD, a high degree of heterogeneity across studies, concerns about the method used to assess telomere length, and the need to adjust for confounding factors in the analyses [44,97]. A more serious concern regarding these two meta-analyses is that the second is a duplicate of the first, drawing on the same pool of five clinical studies [81,83]. Several relevant studies published in the review periods were not included in these meta-analyses for reasons that are unclear, and only some of these were included in the most recent subsequent systematic review. None of these reviews examined associations between telomerase, either in terms of enzyme activity or gene expression, and PTSD.

## 2. Review Process

### 2.1. Objectives and Method of the Current Review

It can be seen from the preceding discussion that there are many lacunae in our understanding of the relationship between PTSD and changes in telomere biology. A preliminary literature search also revealed the presence of several human studies that had been either omitted in the earlier reviews or published subsequent to them. In addition, relevant studies involving animal models of trauma or PTSD have not been evaluated in relation to human research.

The current narrative review was carried out partly to address the above shortcomings and partly to provide an overview of relevant research that has been omitted in the existing systematic reviews. In more specific terms, this review aimed to answer four questions:Is there consistent evidence for an association between PTSD and reduced telomere length? What other variables might affect this association?Is there any evidence for alteration in telomerase activity or expression in patients with PTSD?Is there any evidence of a relationship between traumatic stress or PTSD and telomere length in animal models? Are the findings of animal studies similar to those in humans?Could variations in study quality, methodology, or confounding factors account for the divergent results obtained by different research groups?

### 2.2. Data Retrieval and Synthesis

A synthetic, narrative review method was adopted for two reasons. First, there is marked heterogeneity across study types, and animal and human studies cannot be compared directly. Second, the focus was on addressing specific themes and concepts rather than on quantifying the magnitude of any particular relationship. Standard guidelines for this type of review were followed [98,99]. Relevant literature was retrieved from the PubMed, Scopus, and ScienceDirect databases, with the Google Scholar search engine used as a secondary source for possible grey literature. The search terms “post-traumatic stress disorder”, “posttraumatic stress disorder”, or “PTSD” were used in conjunction with “telomere”, “telomeres”, “telomere length”, “telomerase”, or “telomere shortening”. Both animal and human studies were included. The review process is outlined schematically in Figure 1 below.

A total of 31 relevant reports were retrieved, consisting of 26 human studies and five animal studies. Thirteen of the human studies had been included in at least one of the earlier reviews. Information was retrieved from each paper under the following headings:Study design—cross-sectional or longitudinal, inclusion of a control groupMethod(s) used to assess telomere biologyStudy sample description and sample sizeCovariates or confounding factors studiedResults for primary and secondary outcome variablesStudy quality

As all the studies were cross-sectional biomarker studies, study quality was assessed using the BIOCROSS tool. This instrument consists of ten items covering the five domains of study rationale, methodology, statistical analysis, interpretation, and laboratory techniques [100]. This instrument has been used for the assessment of the quality of biomarker studies in reviews involving patients with other psychiatric disorders [101].

The literature synthesis was conducted in three stages. First, the findings of the thirteen human studies already reviewed elsewhere were summarized. Next, the thirteen human studies not covered in other reviews were described in detail. Finally, animal studies were summarized and possible points of correspondence between animal and human research were examined. On completion of the review, the key findings, the possible mechanisms underlying them, the limitations of the existing data, and the implications of the review findings were summarized.

## 3. Results

### 3.1. Summary and Re-Appraisal of Studies Included in Earlier Systematic Reviews

The three older systematic reviews discussed in Section 1.3.3 included thirteen studies [102,103,104,105,106,107,108,109,110,111,112,113,114]. The detailed characteristics of these studies are summarized in Table 1.

Eight of these studies were conducted in military veterans, and five in civilians exposed to trauma. Except for one study of veterans, all of these were cross-sectional in design. The mean BIOCROSS score for these studies was 14.4 ± 2.5, indicating an above average study quality overall, but with significant variation across reports [100].

Six of the thirteen reports found evidence of a significant reduction in TL associated with PTSD [103,104,105,106,110,112]. Three of these involved military personnel, and three involved civilians exposed to traumatic situations. All but one of these studies adjusted for confounding variables such as age, gender, and body mass index (BMI). Sub-group analyses revealed that childhood trauma might be associated with telomere erosion independent of PTSD, but results were inconsistent [103,106]. The mean BIOCROSS score for these six studies was 13.5, indicating an average study quality.

Of the remaining seven studies, six—five involving military personnel [108,109,111,113,114] and one involving civilians [102]—found no significant association between PTSD and TL. All these studies made appropriate adjustments for confounding factors. Their quality was slightly but not significantly higher than those of the “positive” reports (mean BIOCROSS score = 15.2, indicating above average study quality). In these reports, factors such as childhood trauma, non-specific symptoms of psychological distress, severity of combat exposure, chronic depression, and hostility were linked to reduced TL.

A single study of military veterans found an unexpected association between PTSD and increased TL, which was in sharp contrast to both theoretical expectations and the results obtained by other researchers [107]. This association survived correction for multiple potential confounders. However, there was a significant association between the severity of trauma and TL erosion in the soldiers participating in this study. A noteworthy aspect of this study is that it was longitudinal in design, comparing TL pre- and post-deployment to combat zones. This contrasts with the remaining twelve studies, which were all cross-sectional, and may account for the unexpected result.

Only two of these studies examined possible relationships between the telomerase enzyme and PTSD. In the first, there was no difference in telomerase enzyme activity between male veterans with PTSD and a control group of elderly volunteers [105]. In the second, functional polymorphisms of two genes (*TERT* and *TERC*) encoding components of the telomerase enzyme complex were not associated with PTSD [112].

In summary, less than half of the research reviewed elsewhere found evidence of a link between PTSD and reductions in TL. Factors such as childhood adversity, the severity and duration of trauma, and comorbid mental disorders can significantly influence TL in their own right.

### 3.2. Research on Telomere Length and Telomerase in PTSD Not Reviewed Elsewhere

A total of thirteen published reports, not included in earlier systematic reviews and meta-analyses, have examined the relationship between PTSD and these measures of cellular aging [115,116,117,118,119,120,121,122,123,124,125,126,127]. However, three of these reports all involved analyses of data from a single cohort, with PTSD as only one of many exposures of interest under study [119,120,123]. Results related to PTSD across these four papers were identical and can be combined for the purpose of discussion. Therefore, strictly speaking, there are eleven independent sets of results to be considered in this section. Detailed descriptions of this research are provided in Table 2.

Six of these reports involved military veterans [117,119,120,121,122,123], six involved civilians [115,116,124,125,126,127], and one recruited a mixed sample of civilians and veterans [118]. Two of these studies included a longitudinal component [115,125]. The mean BIOCROSS score for these studies was 16.2, indicating above average study quality. The quality of these reports was significantly better than those included in earlier systematic reviews: no report received a score of less than 14, and their mean BIOCROSS score was significantly higher (independent samples *t* = 2.29, df = 24, *p* = 0.031).

Considering the eleven sets of results, only one found a possible association between PTSD and reduced TL, and even this association was marginal. In a sample of women survivors of sexual assault, re-experiencing symptoms of PTSD were associated with reduced TL at baseline. Other PTSD symptoms were not correlated with TL, and there was no evidence of an association between PTSD and TL one year later [125]. A second study reported a possible longitudinal association between PTSD and reduced TL in civilians, but this was not significant, even at the trend level (*p* = 0.12) [115].

There was no evidence of any association between PTSD and TL, even at a trend level, in six samples—four involving veterans [117,119,120,121,122,123], one involving civilians [127], and one mixed sample [118]. In the remaining three samples, PTSD appeared to be paradoxically associated with increased TL. All these three studies involved civilians who were exposed to significant and sustained trauma—indentured labor, childhood adversity, and exposure to neighborhood violence [116,124,126].

As was the case with the studies discussed in Table 1, a wide range of other variables were associated with alterations in TL in these subjects. Factors associated with reduced TL included depression, generalized anxiety disorder, nicotine dependence, hostility and anger, and certain aspects of military and post-military life. Factors associated with increased TL included positive temperamental traits, self-reported health, and—paradoxically, in some samples—exposure to trauma, violence, or aggression.

Only one of these studies, involving military veterans, examined telomerase enzyme activity in relation to PTSD [121]. In this study, there was no difference in telomerase activity between veterans with and without PTSD. However, telomerase activity was significantly correlated with epigenetic age only in those with PTSD.

In summary, only one of twelve studies included in this part of the review found evidence of a link between PTSD and shortened TL, but this association was of marginal significance. These studies were generally of a higher methodological quality than those included in earlier reviews. Comorbid mental illness and personality traits had a significant influence on the association between TL and PTSD. Positive temperamental traits and good self-reported health had a protective effect on TL. Two studies of civilian populations exposed to severe, chronic trauma reported a paradoxical association between PTSD symptoms and longer TL.

### 3.3. Relationship Between Medical Comorbidities and TL in Patients with PTSD

A limited number of the included studies examined the relationship between TL and medical comorbidities in patients with PTSD. These results are summarized in Table 3. The presence of medical comorbidities was not independently associated with TL in any of these reports. When controlling for the effects of medical comorbidities, three of four reports found that PTSD was still significantly associated with TL shortening [26,35,49,104,110]. There was marked heterogeneity in the assessment of chronic medical conditions: some researchers screened for these conditions using a single general question [124], while others focused on a limited number of specific conditions [110,127].

### 3.4. Other Variables Associated with TL in Patients with PTSD

Psychosocial factors that have been associated with TL in patients with PTSD are summarized in Table 4 below. The most replicated findings suggest that childhood adversity, trauma severity in soldiers exposed to combat, and hostility are significantly associated with reduced TL.

Despite some inconsistency in these results, it was found that factors such as exposure to childhood adversity, the severity of combat trauma exposure in veterans, and self-reported levels of hostility are associated with reduced TL, even after correcting for PTSD diagnosis or severity. A paradoxical effect was noted in two civilian samples, where exposure to violence or trauma seemed to be associated with increased telomere length. Depression was inconsistently associated with TL, though nicotine dependence and general psychological symptoms were linked to reduced TL in veterans.

Higher educational attainment, antidepressant therapy, and self-reported “positive” temperamental traits appeared to be associated with telomere preservation. While the identification of such “protective” factors is important, these results require replication in independent samples.

### 3.5. Animal Research

Research in birds and mammals, not summarized in any previous publication, provides additional insights into the relationship between exposure to severe or traumatic stress and telomere length. These studies are summarized in Table 5.

In four of these five studies, exposure to either a single stressor of high intensity, or to chronic stress over several weeks or months, was associated with reduced TL in mammals [131,132,134] and birds [133]. In two of the mammalian studies, the stressor was of a clearly traumatic nature, involving physical restraint and threats to the animal’s safety. Only one study explicitly examined PTSD-like behaviors in the stress-exposed group. In this study, exposure to trauma was associated not only with telomere shortening, but with increased expression of the telomeric repeat binding factor proteins TRF1 and TRF2, which have a negative regulatory effect on TL [131]. A study of rats exposed to chronic stress found evidence of reduced TL not only in leukocytes, but in neurons of the dentate gyrus of the hippocampus [134]. In meerkat pups exposed to competition for limited food, reduced TL was associated with reduced survival to adulthood [132].

Certain results in the opposite direction were also observed in avian studies. In pied flycatchers exposed to predator threat, adult birds exhibited TL shortening, but nestlings reared at these sites had longer TL [133]. In zebra finches exposed to chronic stress over several months and years, characterized by periods where food was unavailable, there was an increase in TL in middle adulthood, but not in old age. This was associated with reduced mortality at this stage of the life cycle [135].

A further study of interest in this context, though not involving exposure to stress or trauma, was conducted in mice, in which the telomerase gene was knocked out. In old age, these mice showed evidence of reduced TL, reduced hippocampal neurogenesis, neuroinflammation, and impaired memory. However, when these were mated with a mouse strain vulnerable to amyloid plaque accumulation, shortened TL was paradoxically associated with preserved neurogenesis, reduced plaque formation, reduced inflammation, and better learning on experimental tasks [136].

In summary, most of the included studies in animals found an association between traumatic stress, either experimental or naturalistic, and reduced TL. However, the association between animal models of PTSD and TL was examined only in a single study.

## 4. Discussion and Synthesis

### 4.1. Telomere Length

Recent research suggests that the relationship between PTSD and telomere length is not linear. Though initial meta-analyses suggested that PTSD was significantly associated with telomere shortening or erosion, results from the past decade have yielded more mixed results that do not lend themselves to simple explanations. These results can be summarized as follows:Depending on the population studied and the methods used, PTSD may be associated with either an increase, a decrease, or no significant change in TL.There does not appear to be any significant correlation between PTSD symptom severity, as measured using standardized instruments, and TL.Other psychological variables may have a significant effect on TL in persons with PTSD.There is no clear evidence that TL in PTSD predicts physical health outcomes, such as chronic medical illnesses or neurodegenerative disorders.

Several factors may account for this marked variability in results across studies.

#### 4.1.1. Methodological Issues

There are various methods of measuring TL, but not all of them correlate well with physiological outcomes. For example, the shortest TL, rather than the average TL, is a reliable biomarker of the onset of cellular aging and cessation of cell division [44,137,138]. Moreover, variations in the cell type used to estimate TL, the equipment used for estimation, or time of sample processing may also lead to variations in this parameter. This was illustrated in a study of PTSD in veterans, in which TL varied significantly across the two batches of samples processed [43]. Studies with small sample sizes, with few participants having PTSD, may lack the statistical power to identify meaningful differences in TL between cases and controls. On the other hand, studies that do not control for possible confounders may over- or underestimate the association between PTSD and changes in TL. Other sources of bias, including a low participation rate or a lack of precision in defining criteria for cases and controls, are also important in biomarker research in general and may have affected study outcomes either positively or negatively in some cases [100].

#### 4.1.2. The Role of Confounding Factors

When evaluating the results of the research cited above, it is sometimes assumed that TL is a common biomarker for both depression and PTSD [139]. As has been noted in the first meta-analysis by Darrow et al., reduced TL has been observed in a number of mental disorders, including anxiety disorders, mood disorders, and schizophrenia. Sub-group analyses revealed a relatively weaker effect for severe mental disorders—schizophrenia and bipolar disorder—and a larger effect for “common” or stress-related disorders such as depression, anxiety disorders, and PTSD [81]. Some mental disorders, such as eating disorders, do not seem to be associated with TL shortening [140]. For this reason, some experts have suggested in general [141]. The associations between stress or stress-related illnesses and TL are modest in magnitude and are influenced by specific variables such as gender, socioeconomic status, the severity of the stressor, and its consequences [142,143].

Even if telomere shortening is associated with PTSD, its severity appears to be influenced by several factors, as summarized in Table 4. Of these potential confounders, the one that has been studied most extensively is childhood adversity. Four of six published reports found that adverse childhood experiences (ACE) were associated with reduced TL in patients with a diagnosis of PTSD. Moreover, there appears to be a “dose-response” relationship for this association: a single type of childhood abuse or neglect was not associated with significant TL shortening, while those with two, three, or more such types of adversity showed an ordinal decrease in TL [103]. Similar results have been obtained in persons without a diagnosis of PTSD, suggesting that this effect is specific to ACE regardless of the presence of a given mental disorder [120].

Combat trauma is one of the main causes of PTSD in military personnel, particularly those deployed in zones of armed conflict, and more severe or prolonged trauma is associated with higher levels of PTSD symptoms [144]. Two studies of veterans with PTSD found an independent association between combat trauma severity and reduced TL [107,114]. Combat trauma is a complex form of traumatic stress that involves actual and threatened physical injury or death, either of oneself or one’s comrades, as well as moral injury (e.g., guilt over causing injury or death to opposing combatants) [145,146]. This could explain its observed, specific relationship with telomere shortening.

Hostility can be understood as a “cognitive component of anger”, characterized by cynicism and a tendency to interpret others’ actions as harmful or hurtful to oneself. This leads to emotions of anger and disgust towards others [147]. Over 30% of those suffering from PTSD, whether civilian or military, have high levels of hostility. These patients tend to have higher functional impairment, more comorbid depression, poorer social support, and an increased risk of suicidal behavior [147,148]. Given the existence of a replicated association between hostility and reduced TL in military personnel with PTSD [113,117], the mechanisms through which this variable influences telomere biology require further study.

Synergy between these factors in influencing TL is also possible. In a longitudinal study of over 1200 veterans, it was found that childhood adversity influenced the risk of developing PTSD after combat trauma through a hypothesized process of “stress sensitization” [149]. Such a process could lead to exaggerated stress responses that accelerate telomere loss or erosion.

An exhaustive review of these and other factors linked to TL in patients with PTSD is beyond the scope of this paper. The key issue at hand is that they need to be considered as confounding or interacting variables when analyzing data on the link between PTSD and telomere erosion. In addition, there are several factors that have been shown to influence TL, independent of any medical or psychiatric diagnosis. Foremost among these are genetic influences, which may account for up to 60–80% of the variation in TL between subjects. It is a significant limitation of the existing research on PTSD and TL that it has not taken the role of genetic variations in TL into account.

### 4.2. Insights from Animal Research

In contrast to the mixed results obtained in human studies, most animal studies examining the effects of chronic stress or trauma have found evidence of telomere shortening. This effect is so marked that telomere attrition has been put forward as a candidate biomarker of animal welfare, especially in the context of animal husbandry and laboratory research [150]. This consistency may result from the greater ease of standardizing exposures and minimizing confounders when environmental conditions can be easily controlled.

In the one animal study in which stress was associated with increased rather than decreased TL, the experimental stressor was of a more chronic and naturalistic nature, involving periodic food reduction without a threat to the animal’s life or physical safety [135]. Three of the human studies that found evidence of increased TL involved individuals exposed to chronic stress, both traumatic and non-traumatic, over prolonged periods. In these studies, TL was positively correlated with measures of stress exposure, suggesting that under certain circumstances, stress may lead to the development of a form of resilience at the molecular or cellular level [116,124,126]. It is possible, as the authors of one of these papers have speculated, that chronic stress may sometimes lead to “the induction of protective homoeostatic mechanisms”, both in animals and in humans [126]. Such a mechanism of cellular resilience or “antifragility” may protect against age-related diseases and improve survival, as was observed in the animal model, but this possibility is yet to be verified in humans [151].

### 4.3. Relationship Between TL and Health Outcomes in PTSD

There is evidence from clinical research that PTSD is associated with multiple medical conditions related to aging, as well as with dementia [39,42,152,153]. In this context, it would be relevant to know whether the occurrence of these disorders in patients with PTSD is associated with telomere erosion or altered telomerase activity. As per the evidence reviewed in Table 3, there does not appear to be a specific association between the presence of any medical illness and TL in these patients. Moreover, the association between PTSD and TL is not consistently influenced by these comorbidities. To date, there is no research in this field examining a specific link between TL and neurodegeneration or cognition in PTSD. On a related note, PTSD does not appear to be associated with a significantly elevated risk of cancers. However, there is evidence of a link between PTSD and ovarian carcinoma, which is characterized by increased rather than decreased telomerase activity [154,155].

### 4.4. Telomerase

When compared with the number of studies examining TL in PTSD, there is little relevant research on telomerase in relation to this disorder. Two studies, both involving military veterans exposed to combat, found no difference in telomerase activity between patients and controls. There is some evidence linking chronic stress to reduced telomerase activity, but this relationship is complex. For example, emotional abuse in childhood has been associated with an increase in telomerase activity, and similar findings have been observed in patients with depression [156,157]. It has also been observed that the leukocytes of individuals experiencing stress in the absence of social support or other protective factors show a paradoxical combination of short telomeres and increased telomerase activity [158]. This may reflect the involvement of other cellular mechanisms causing telomere shortening, even in the presence of normal or increased telomerase activity. For example, in an animal model of PTSD, shortened TL was associated with increased expression of TRF1 and TRF2 [131]. These proteins are components of the six-protein shelterin complex that maintains telomere integrity. Increased expression of TRF1 has been found to reduce telomere length, and it has been suggested that this occurs through binding to the ends of telomeres, preventing the telomerase enzyme complex from accessing them [159].

Though PTSD is associated with an increased risk of subsequent neurodegenerative disorders, it is not clear whether telomerase activity is related to this outcome. Animal research suggests that telomerase enzyme activity has variable effects on the risk of neurodegeneration or dementia. In “normal” mice, reduced telomerase appears to promote neuronal loss and inflammation, but in those with a genetic risk of Alzheimer’s disease, reduced telomerase appears to be protective, perhaps by promoting the senescence and death of cells involved in amyloid deposition [136].

From the limited evidence available, it can be tentatively concluded that telomerase activity is not significantly altered in PTSD and does not appear to link this disorder to subsequent health outcomes.

### 4.5. Clinical and Research Implications

A careful review of the available evidence, including research that has been omitted from earlier systematic reviews, suggests that there is no consistent association between PTSD and reduced TL or telomere erosion. In fact, there is as much evidence for increased as for reduced TL in this disorder, and the majority of studies found no association in either direction. In addition, there is no data to suggest that reduced TL is associated with the presence of specific medical or neurological comorbidities in patients with PTSD. Overall, there are no consistent results to suggest that TL is a biomarker of PTSD or a biomarker of premature aging in PTSD. Reliable evidence for a link between telomerase activity and PTSD is scanty, but the few published results suggest that there is no association between them. It is unlikely that pharmacological manipulation of telomerase is a viable therapeutic strategy in this condition or its medical comorbidities, despite its promise as a treatment for aging-related or neurodegenerative diseases in general [160].

The lack of a consistent link between PTSD and TL shortening does not invalidate the hypothesis of accelerated aging in PTSD. Due to the large number of factors that can influence TL, including a substantial genetic component, a more plausible conclusion is that *TL is neither sensitive nor specific enough to act as a marker of cellular aging in PTSD*. It is unlikely that this limitation can be circumvented simply through methodological refinements or better statistical modeling of confounders. Instead, the findings of this review could serve to draw attention away from TL and towards other potential markers of aging in PTSD. These include altered DNA methylation patterns [161,162] and age-related markers of inflammation such as IL-6 and CRP [38,163]. If a specific link between PTSD and neurodegeneration is being investigated, markers of brain aging, such as volume loss in key brain regions or evidence of cognitive decline on longitudinal assessment using standardized tests, may provide evidence to support this link and elucidate the mechanisms underlying it [163,164].

## 5. Conclusions

Though initial reports suggested that post-traumatic stress disorder was associated with telomere shortening, a marker of cellular aging, subsequent research has failed to confirm this possibility. Telomere length in PTSD is highly variable and influenced by a number of demographic, trauma-specific, psychological, and social variables. Furthermore, there is no evidence to support earlier speculations on reduced telomerase activity in PTSD leading to telomere shortening. Some of this variability may be due to methodological and statistical limitations. It is also possible that telomere length may not be a sufficiently specific marker of accelerated aging in patients with PTSD. Rather than attempting to conduct further research on TL, it may prove more fruitful to study other markers of cellular aging in this patient population. These may provide more sensitive and specific tests of the “accelerated aging” hypothesis in PTSD, both in general and in specific tissues such as the brain.

## Figures and Tables

**Figure 1 biomedicines-13-00507-f001:**
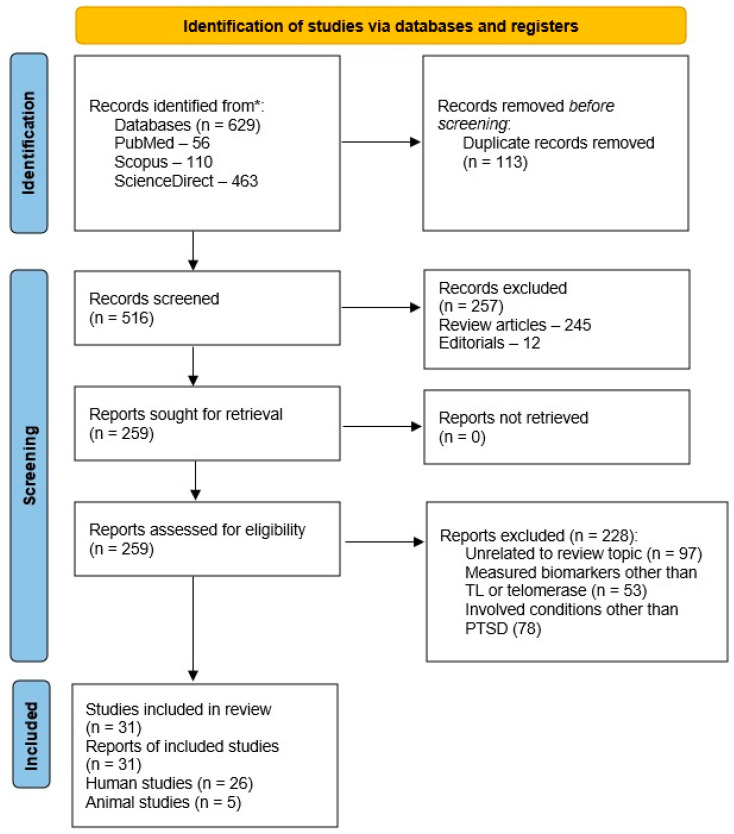
Schematic depiction of the selection and retrieval of papers for this study. Abbreviations: PTSD, post-traumatic stress disorder; TL, telomere length. * Three databases were consulted for this review. See the text for the complete details.

**Table 1 biomedicines-13-00507-t001:** Studies of telomere shortening in post-traumatic stress disorder included in earlier (2017–2021) systematic reviews.

Study and Country of Origin	Study Design	Sample Size and Characteristics	Covariate(s) Studied (If Applicable)	Method(s) Used to Assess Telomere Biology	Results—Main Analysis	Results—Subgroups and Other Outcomes (If Analyzed)	Study Quality (BIOCROSS Score)
Malan et al., 2011 [102] South Africa	Cross-sectional	Female survivors of rape (*n* = 64; 31 with MDD, 9 with PTSD). Diagnoses made using DSM-IV criteria.	Age, ethnicity, education.	Relative LTL using qPCR	Marginal association (*p* = 0.05) between PTSD and reduced relative LTL, even after adjusting for age.	No association between relative LTL and MDD or self-reported resilience.	15
O’Donovan et al., 2011 [103] United States of America	Cross-sectional, case-control	Adults with PTSD related to childhood trauma (*n =* 43) and healthy controls (*n =* 47). Diagnoses made using DSM-IV criteria.	Age, depression, number of types of childhood trauma.	Relative LTL using qPCR	Significantly reduced LTL in PTSD vs. controls after age adjustment (*F* = 3.29, *p* = 0.03); significant negative correlation between childhood trauma and LTL (partial *r* = −0.27, *p* = 0.005).	PTSD associated with reduced LTL only in those with multiple types of childhood trauma.	14
Ladwig et al., 2013 [104] Germany	Cross-sectional	Adults from the general population (*n* = 3000; 51 with PTSD, 262 with “partial PTSD”). Diagnoses made using PDS and IES.	Age, gender, relationship status, education, BMI, smoking, alcohol use, physical activity, depression, medical comorbidities (dyslipidemia, systemic hypertension, diabetes mellitus, cerebrovascular or cardiovascular disease, or cancer).	Average LTL using qPCR	Both “partial” and full PTSD associated with reduced TL after adjusting for age, gender, and BMI, with a small effect size (β = −0.05 to −0.11, *p* = 0.007 to 0.014).	Unadjusted average TL not significantly different across groups; significant correlation between age and PTSD diagnosis.	13
Jergovic et al., 2014 [105] Croatia	Cross-sectional, case-control	Male military veterans with combat-related PTSD (*n* = 30); age-matched healthy controls (*n* = 17); elderly volunteers without PTSD (*n* = 15; 13 female, 2 male). Diagnoses made using ICD-10 criteria.	Age, BMI, smoking, alcohol use, medication intake.	Relative LTL using multiplex qPCR	Both PTSD and elderly volunteers had shorter average TL than healthy controls; no difference in telomerase activity between PTSD and healthy controls.	Lower telomerase in elderly volunteers than healthy controls.	14
Zhang et al., 2014 [106] United States of America	Cross-sectional, case-control	Military veterans (*n =* 650; 84 with PTSD). Diagnoses made using DSM-IV criteria and PCL score ≥ 50.	Age, exposure to stressful life events.	Relative LTL using real-time PCR	PTSD associated with reduced relative LTL even after adjusting for age (*p* < 0.01).	Stressful life events associated with increased LTL in non-PTSD subjects. Age negatively correlated with LTL in non-PTSD but not in PTSD subjects. No association between childhood trauma and LTL.	10
Boks et al., 2015 [107] The Netherlands	Longitudinal	Male soldiers exposed to combat trauma (*n* = 96; 32 with significant PTSD symptoms). PTSD symptoms measured using SRIP.	Age, gender, smoking, alcohol use, BMI, medication intake.	Duplicate determination of TL using qPCR	PTSD associated with increased TL compared to pre-deployment value (*p* = 0.018), even after adjustment for possible confounders.	Severity of trauma exposure associated with reduced TL compared to pre-deployment value.	15
Bersani et al., 2016 [108] United States of America	Cross-sectional	Male military veterans exposed to combat (*n* = 76; 18 with PTSD, 17 with both PTSD and MDD). Diagnoses made using DSM-IV criteria. PTSD symptoms measured using CAPS.	Age, ethnicity, body mass index, smoking, perceived stress, general psychological symptoms, depression, antidepressant intake.	Relative LTL using qPCR	No significant association between either PTSD diagnosis or PTSD symptom severity and TL.	TL significantly and negatively associated with childhood trauma, general psychological symptoms, and perceived stress. No association between MDD and TL.	15
Kim et al., 2017 [109] South Korea	Cross-sectional	Male military veterans with (*n* = 122) and without (*n* = 120) PTSD. Diagnoses made using DSM-IV-TR criteria. PTSD symptoms measured using CAPS.	Age, education, socioeconomic status, smoking, alcohol use, antidepressant intake.	Relative LTL using qPCR	No difference in TL between PTSD and non-PTSD groups.	In a sub-group of 45 veterans exposed to “severe combat”, PTSD modestly associated with reduced TL (*p* = 0.029). Age negatively correlated, and education and antidepressant use positively correlated with TL in this group.	17
Roberts et al., 2017 [110] United States of America	Cross-sectional	Civilian female nurses (*n* = 116; 25 with PTSD, 66 with “subclinical” PTSD). Diagnoses made using DSM-IV criteria.	Age, BMI, smoking, alcohol use, dietary pattern, physical activity, type of trauma, antidepressant intake, medical comorbidities (dyslipidemia and systemic hypertension).	Relative LTL using modified real-time qPCR	Significant but modest association between PTSD and shortened TL (β = −0.11, *p* < 0.05) even after adjusting for confounders. Non-significant association between subclinical PTSD and shortened TL.	Type of trauma not associated with TL and did not mediate the association between PTSD and TL.	19
Solomon et al., 2017 [111] Israel	Retrospective cohort	Former soldiers captured as prisoners of war (*n* = 90) evaluated up to 42 years after repatriation and healthy controls with past military service (*n* = 79). Diagnoses made using DSM-IV-TR criteria.	Age, BMI, depression	Average LTL using Southern blot	No significant effect of past or current PTSD on TL.	Shorter TL in prisoners compared to controls. Chronic depression associated with reduced TL in prisoner group.	15
Avetyan et al., 2019 [112] Armenia	Cross-sectional, case-control	Male military veterans with PTSD (*n* = 41) and matched healthy controls (*n* = 49). Diagnoses made using DSM-IV-TR criteria. PTSD symptoms measured using CAPS.	Not mentioned.	Relative LTL using multiplex qPCR Genotyping of *TERC* and *TERT* using PCR-SSP	Average LTL reduced approximately 1.5-fold in PTSD patients compared with controls (*p* = 0.03). No correlation between CAPS score and LTL.	*TERT* rs2736100 T allele 1.6 times more common in PTSD than in control group. No association between *TERT* genotype and CAPS score or LTL. No association between *TERC* genotype and PTSD.	11
Zhang et al., 2019 [113] United States of America	Cross-sectional	Military veterans with combat exposure (*n* = 474). PTSD symptoms measured using PCL.	Age, gender, education, self-reported hostility.	Relative average LTL using real-time PCR	PTSD not significantly correlated with LTL overall.	PTSD associated with higher hostility scores. Hostility associated with shorter LTL in veterans with PTSD. Hostility, especially hostile impulses, negatively associated with LTL.	12
Kang et al., 2020 [114] United States of America	Cross-sectional	Male military veterans with (*n* = 102) or without (*n* = 111) PTSD. Diagnoses made using DSM-IV criteria. PTSD symptoms measured using CAPS.	Age, childhood trauma, perceived stress, self-reported positive and negative mood states, depression, smoking, alcohol use.	Relative LTL using qPCR	No significant association between PTSD and TL.	PTSD associated with shorter TL in veterans with high, but not low trauma exposure. Urinary norepinephrine negatively correlated with TL.	17

Abbreviations: BMI, body mass index; CAPS, Clinician-Administered PTSD Scale; DSM, Diagnostic and Statistical Manual of Mental Disorders; IES, Impact of Event Scale; LTL, leukocyte telomere length; MDD, major depressive disorder; PCL, PTSD Checklist; PCR, polymerase chain reaction; PCR-SSP, polymerase chain reaction with sequence-specific primers; PDS, Posttraumatic Diagnostic Scale; PTSD, post-traumatic stress disorder; qPCR, quantitative polymerase chain reaction; SRIP, Self-Rating Inventory for PTSD; *TERC*, telomerase RNA component gene; *TERT*, telomerase reverse transcriptase gene; TL, telomere length.

**Table 2 biomedicines-13-00507-t002:** Studies of telomere length and telomerase in post-traumatic stress disorder not included in other reviews.

Study and Country of Origin	Study Design	Sample Size and Characteristics	Covariate(s) Studied (If Applicable)	Method(s) Used to Assess Telomere Biology	Results—Main Analysis	Results—Subgroups and Other Outcomes (If Analyzed)	Study Quality (BIOCROSS Score)
Shalev et al., 2014 [115] New Zealand	Longitudinal	Adults from the general population (*n* = 758, 113 with MDD, 51 with GAD, 32 with PTSD). Diagnoses made using DSM-III-R or DSM-IV criteria.	Socioeconomic status, childhood abuse, smoking, substance dependence, psychiatric medication intake, self-reported physical health.	Average LTL using qPCR	Non-significant association between PTSD in men at age 26–38 and LTL erosion at age 38 (β = −0.07, *p* = 0.12).	Significant LTL erosion in men, but not women, with MDD or GAD.	17
Kuffer et al., 2016 [116] Switzerland	Retrospective cohort	Elderly former indentured laborers (*n* = 62, 21 with “partial or full” PTSD) and elderly healthy controls (*n* = 58). PTSD symptoms measured using SSS.	Age, gender, education, self-reported financial status, depression.	Mean buccal TL using qPCR	Significantly longer buccal TL in persons with PTSD than in controls (*p* = 0.04). No difference in buccal TL between laborers with and without PTSD.	Childhood trauma marginally associated with longer buccal TL (*p* = 0.05).	16
Watkins et al., 2016 [117] United States of America	Cross-sectional	Military veterans (*n* = 468; 83 with lifetime PTSD or MDD). Diagnoses made using DSM-IV criteria. PTSD symptoms measured using PCL.	Age, gender, marital status, employment status, lifetime traumatic events, depression, substance dependence, physical health, current level of psychological distress, self-reported hostility.	Average LTL using qPCR	No association between PTSD and shortened TL in univariate or multivariate analyses.	TL shortening significantly associated with nicotine dependence and self-reported hostility and anger. No association between MDD and TL.	17
Connolly et al., 2018 [118] United States of America	Cross-sectional	Trauma-exposed adults (*n* = 453; 314 veterans, 139 civilians; 177 with PTSD). Diagnoses made using DSM-IV criteria. PTSD symptoms measured using CAPS.	Age, gender, lifetime traumatic events, self-reported temperamental traits, medical comorbidities (dyslipidemia, systemic hypertension, diabetes mellitus or cardiovascular disease).	Average relative LTL using qPCR	No significant association between PTSD and TL when controlling for age.	PTSD negatively associated with TL in subjects older than 55. Temperamental traits of positive emotionality and drive to achieve positively associated with TL.	18
Stein et al., 2018 [119] Israel	Cross-sectional	Former soldiers captured as prisoners of war (*n* = 99) evaluated 42 years after repatriation. Diagnoses made using DSM-IV criteria. PTSD symptoms measured using PTSD-I.	Age, BMI, smoking, substance abuse, physical activity, characteristics of captivity, post-captivity life events.	Average LTL using Southern blot	No significant association between either PTSD diagnosis or PTSD symptom score and TL.	TL significantly and negatively associated with aspects of captivity (solitary confinement) and post-captivity life (loss of family after release, being accused after repatriation, loneliness).	15
Tsur et al., 2018 [120] Israel	Cross-sectional	Former soldiers captured as prisoners of war (*n* = 88) evaluated 42 years after repatriation. Diagnoses made using DSM-IV criteria. PTSD symptoms measured using PTSD-I.	Age, BMI, smoking, substance abuse, medical comorbidities, self-rated health.	Average LTL using Southern blot	No significant association between PTSD and TL in univariate or multivariate analyses.	Self-rated health positively correlated with TL. PTSD negatively associated with self-rated health.	14
Verhoeven et al., 2018 [121] United States of America	Cross-sectional	Male military veterans with combat exposure (*n* = 160; 79 with PTSD). Diagnoses made using DSM-IV criteria. PTSD symptoms measured using CAPS.	Age, ethnicity, BMI, smoking, alcohol use, childhood trauma, number of tours of military duty, antidepressant intake.	Relative LTL using qPCR	No significant association between PTSD and TL or telomerase activity after adjusting for age.	Telomerase activity negatively associated with epigenetic age in PTSD group only.	17
Wolf et al., 2019 [122] United States of America	Cross-sectional	Military veterans of Caucasian ethnicity with combat exposure (*n* = 309; 198 with PTSD). Diagnoses made using DSM-IV criteria. PTSD symptoms measured using CAPS.	Age, gender, traumatic life events, self-reported pain, self-reported sleep quality.	Relative average LTL using real-time qPCR	No significant association between PTSD symptom severity and TL.	Age significantly associated with reduced TL. No associations between functional polymorphisms of *KL* gene and TL.	16
Ein-Dor et al., 2020 [123] Israel	Cross-sectional	Former soldiers captured as prisoners of war (*n* = 88) assessed up to 42 years after repatriation. PTSD symptoms measured using PTSD-I.	Age, attachment orientation, partner’s attachment orientation, BMI, smoking, substance abuse, physical activity, medication intake.	Average LTL using Southern blot	No significant association between PTSD and TL.	TL negatively associated with subjects’ attachment avoidance and their spouses’ attachment anxiety. Spousal attachment avoidance associated with longer TL.	15
Burgin et al., 2020 [124] Switzerland	Cross-sectional	Young adults leaving residential care (*n* = 130; 29 with lifetime PTSD).	Age, gender, childhood adversity, lifetime traumatic events, migrant background, internalizing symptoms.	Average LTL using qPCR	Lifetime PTSD associated with longer TL (*p* < 0.001).	Exposure to traumatic stress associated with longer TL.	16
Carvalho et al., 2022 [125] Brazil	Longitudinal	Women with PTSD following sexual assault followed up for 1 year (*n* = 64); healthy controls (*n* = 60). Diagnoses made using DSM-III/DSM-IV criteria. PTSD symptoms measured using CAPS.	Age, income, alcohol use.	Relative LTL using multiplex real-time qPCR	Baseline PTSD re-experiencing symptoms associated with reduced relative LTL (β = −0.02, *p* = 0.02). No association between other PTSD symptom dimensions and LTL.	No longitudinal association between PTSD and relative LTL. Trend towards shorter LTL in those whose PTSD had remitted at follow-up.	15
Womersley et al., 2022 [126] South Africa	Cross-sectional	Black men residing in areas with high rates of community violence (*n* = 290; 138 with high PTSD symptoms). PTSD symptoms measured using PSS-I.	Age, education, self-reported appetitive aggression.	Relative LTL using qPCR	PTSD symptom severity positively correlated with relative TL (*p* = 0.016).	Among PTSD symptoms, flashbacks, and emotional numbing positively associated with relative TL. Measures of aggression and of exposure to community violence positively correlated with relative TL.	16
Ratanatharathorn et al., 2023 [127]	Cross-sectional	Female nurses (*n* = 1868; 834 with lifetime trauma but no diagnosis; 238 with PTSD alone; 327 with MDD; 175 with both PTSD and MDD). Diagnoses made using DSM-5 criteria. PTSD symptoms measured using SSS.	Age, ethnicity, BMI, smoking, alcohol use, physical activity, dietary pattern, medical comorbidities (dyslipidemia, diabetes mellitus), cholesterol-lowering medication intake.	Relative LTL using modified real-time qPCR	No significant association between PTSD and LTL.	Neither trauma nor MDD associated with LTL. Comorbid PTSD and MDD associated with significantly shorter LTL, particularly in those with higher PTSD symptom scores.	18

Abbreviations: BMI, body mass index; CAPS, Clinician-Administered PTSD Scale; DSM, Diagnostic and Statistical Manual of Mental Disorders; GAD, generalized anxiety disorder; *KL,* Klotho gene; LTL, leukocyte telomere length; MDD, major depressive disorder; PCR, polymerase chain reaction; PSS-I, PTSD Symptom Scale—Interview; PTSD, post-traumatic stress disorder; PTSD-I, PTSD Inventory; qPCR, quantitative polymerase chain reaction; SSS, Short Screening Scale for PTSD; TL, telomere length.

**Table 3 biomedicines-13-00507-t003:** Associations between medical comorbidities and telomere length in post-traumatic stress disorder.

Study	Significant Result(s)
Ladwig et al., 2013 [104]	Chronic medical illnesses present in 20–23% of those with PTSD. Association between PTSD and TL shortening significant, even after adjusting for medical comorbidities.
Bersani et al., 2015 [108]	Negative correlation between PTSD symptom severity and TL no longer significant after adjusting for medical comorbidity.
Watkins et al., 2016 [117]	Neither number of chronic medical illnesses nor PTSD significantly associated with TL.
Roberts et al., 2017 [110]	Hypercholesterolemia present in 12% and systemic hypertension in 8% of participants with PTSD. Association between PTSD and TL shortening significant, even after adjusting for these comorbidities.
Tsur et al., 2018 [120]	No significant correlation between the presence of medical comorbidities (cardiovascular, pulmonary, neurological, renal, or metabolic) and TL.
Burgin et al., 2022 [124]	Presence of “any acute or chronic illness” associated with longer TL in only one of the four multivariate models.
Ratanatharathorn et al., 2022 [127]	Systemic hypertension, type II diabetes mellitus, or hypercholesterolemia present in 10–11% of participants with PTSD and 21–27% of participants with comorbid PTSD/MDD. Association between comorbid PTSD/MDD and TL shortening significant, even after adjusting for these comorbidities.

Abbreviations: MDD, major depressive disorder; PTSD, post-traumatic stress disorder; TL, telomere length.

**Table 4 biomedicines-13-00507-t004:** Variables associated with altered telomere length in patients with post-traumatic stress disorder.

Variable	Effect on Telomere Length	References
**Demographic**		
Gender (male)	↓	Shalev et al., 2014 [115]
Education	↑	Kim et al., 2017 [109]
**Stress and trauma-related**		
Severity of trauma	↓ (military) ↑ (civilian)	Boks et al., 2015 [107]; Kang et al., 2020 [114] Burgin et al., 2020 [124]
Exposure to community violence	↑	Womersley et al., 2022 [126]
Self-reported stress	⇔	Bersani et al., 2016 [108]; Kang et al., 2020 [114]
Childhood adversity	↓ (in four of six studies)	O’Donovan et al., 2011 [103]; Bersani et al., 2016 [108]; Kuffer et al., 2016 [116]; Kang et al., 2020 [114]; Chen et al., 2019 [128]; Zhou et al., 2023 [129]
**Mental health-related**		
Specific PTSD symptoms of flashbacks and numbing	↑	Womersley et al., 2022 [126]
Depression	⇔	Solomon et al., 2017 [36]; Ratanatharathorn et al., 2022 [49]
Nicotine dependence	↓	Watkins et al., 2016 [117]
General psychological symptoms	↓	Bersani et al., 2016 [108]
Antidepressant treatment	↑	Kim et al., 2017 [109]
**Psychological and behavioral**		
Resilience	⇔	Malan et al., 2011 [102]; Zerach et al., 2020 [130]
Hostility	↓↓	Watkins et al., 2016 [117]; Zhang et al., 2019 [113]
Aggression	↑	Womersley et al., 2022 [126]
Attachment avoidance	↓	Ein-Dor et al., 2020 [123]
Positive emotionality	↑	Connolly et al., 2018 [118]
Drive to achieve	↑	Connolly et al., 2018 [118]
**Others**		
Self-rated health	↑	Tsur et al., 2018 [120]
Spouse’s attachment anxiety	↓	Ein-Dor et al., 2020 [123]
Spouse’s attachment avoidance	↑	Ein-Dor et al., 2020 [123]

Abbreviations: TL, telomere length; ↓, TL reduced in a single study or in most studies; ↓↓, TL consistently reduced across multiple studies; ↑, TL increased in a single study; ⇔, inconclusive or insufficient evidence across studies.

**Table 5 biomedicines-13-00507-t005:** Animal studies of telomere length and telomerase in relation to severe stress.

Study	Species and Description	Stress Exposure	Results
Dong et al., 2016 [131]	Male Wistar rats	Single Prolonged Stress (SPS)—forced restraint for 2 h followed by forced swimming for 20 min.	SPS associated with accelerated telomere shortening. PTSD-like behaviors due to SPS associated with increased expression of TRF1 and TRF2, which are negative regulators of TL.
Cram et al., 2017 [132]	Wild meerkat pups	Competition from other pups in the group for limited nutritional resources.	High competition for nutrition associated with reduced TL. This effect was reversed when maternal nutrition was improved. Pup TL positively associated with survival to adulthood.
Karkkainen et al., 2019 [133]	Pied flycatchers	Predator threat from pygmy owls.	Shortened TL in parent birds at owl-exposed sites. Increased TL in chicks reared at owl-exposed sites.
Lee et al., 2021 [134]	Male Sprague-Dawley rats	Chronic Variable Stress—physical restraint, cold exposure, and cage shaking twice a day at irregular times, along with social crowding, restricted food, wet bedding, and light exposure during the night for 3 weeks.	Significant reduction in TL in lymphocytes and hippocampal neurons of stressed rats.
Marasco et al., 2021 [135]	Female zebra finches	Exposure to “challenging” environment—unavailability of food for one-third of the day—at random for four days a week, starting from the age of 5 months.	Stress-exposed birds had reduced telomere loss in middle adulthood. Reduced mortality in middle age, but not old age, in stress-exposed birds.

Abbreviations: TL, telomere length; TRF, telomeric repeat binding factor.

## Data Availability

No new data was generated for the purpose of this paper.

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
