# Peer review of "Telomere Dynamics in Post-Traumatic Stress Disorder: A Critical Synthesis"

_biomedicines, 2025, doi:10.3390/biomedicines13020507_

Round 1
Reviewer 1 Report
Comments and Suggestions for Authors
28 August 2024
The review report on the manuscript, titled ‘Telomere length and telomerase in post-traumatic stress disorder: an update and a reappraisal of the evidence’ by Rajkumar, RP submitted to Biomedicines
Manuscript ID: biomedicines-3193437
Dear Author,
There is still inconclusive evidence about the relationship between post-traumatic stress disorder (PTSD) and telomere length or telomerase activity, emphasizing the need for more rigorous and methodologically sound research into these potential biomarkers of accelerated aging. The current challenges include high variability in telomere length findings across studies, as well as a lack of consistent evidence linking PTSD to altered telomerase activity. The reliability of telomere length (TL) as a biomarker for PTSD remains uncertain due to methodological differences, small sample sizes, and confounding factors. In this manuscript, entitled ‘Telomere length and telomerase in post-traumatic stress disorder: an update and a reappraisal of the evidence,’ Rajkumar explores the relationship between PTSD and cellular aging, focusing on TL and telomerase activity as potential biomarkers of accelerated aging.
The strength of this study lies in its comprehensive and critical review of recent clinical and translational research on the relationship between PTSD, telomere length, and telomerase activity. By systematically analyzing diverse studies, the manuscript highlights the complexities and methodological challenges in this field, offering valuable insights into why previous findings have been inconsistent.
In general, I am intrigued by the concept of this research paper, and the author's fascinating observations on this current topic is potentially of interest to Biomedicines readers. To improve the manuscript's quality before publication, the author should include critical evidence and comments that reinforce the sufficiency and readability of their arguments. Finally, I suggest that the author considers my comments and suggestions before publishing this research paper.
Comments:
1. First, I would like the author to clarify the type of review article, such as systematic, scoping, or synthetic, and declare it in the title, abstract and the objectives of introduction. Then, I would like the author to make sure there are all elements necessary for a certain type of review paper by using the checklist [1–3].
2. Title: Please provide a concise and informative title that accurately reflects the key message of this study, as this is the most essential aspect of the manuscript. Please consider: "Unraveling the Telomere Puzzle: PTSD and the Complexities of Cellular Aging"; "PTSD and Accelerated Aging: A Critical Reappraisal of Telomere Dynamics".
3. Abstract: The abstract is generally well written, but please reconsider the following guidelines: I recommend that the author presents the background, methods, results, and conclusion in a proportional order within 200 words without subheadings. The general background (one to two sentences), specific background (two to three sentences), and current issue addressed in this study (one sentence) should all be included in the background before moving on to the objectives. In this subsection, I would like the author to provide background information, a problem statement, and their reasoning for breaking off. The results section ends with a phrase that places this subsection in a broader context. The conclusion should start with a single sentence that summarizes the main message, such as "Here we show." The author should describe the potential and progress of this study in the field in the first sentence of the conclusion, followed by two to three sentences that provide a broader perspective that any scientist can comprehend. Please refrain from presenting statistical details in the abstract.
4. Keywords: Please include ten keywords from Medical Subject Headings (MeSH) in the title and the first two sentences of the abstract.
5. I highly recommend presenting an informative graphical or video abstract.
6. Introduction: This section is well written, but please consider expanding this section according to the following guidelines: I recommend that the author presents this section before presenting the details of the following sections, incorporating several paragraphs totaling approximately 1000 words, to provide a comprehensive introduction to the main constructs of this study. This introduction should be understandable to readers from all disciplines and serve to establish the purpose of the current research as well as the specific intent of this study. The author is advised to begin with a general background, then move on to a more specific background, and the current issue addressed in this study, leading to the objectives. The manuscript's objectives should be reached through a coherent and logical progression.
7. Methods: Please present the method section if this manuscript is systematic, scoping, or synthetic review. This section should start with a brief introduction and then cites additional references to ensure the reliability and integrity of the evidence in the author’s study design and methodology. This section's key elements include a PRISMA flow chart.
8. Results: Please present the result section if this manuscript is intended to be systematic, scoping, or synthetic review. I recommend including a paragraph that provides a summary of the findings as the section's conclusion. This section's key elements include a risk of bias assessment.
9. Discussion: I would like the author to present the paragraphs of the discussion section totaling around 1,500 words, beginning with an introductory paragraph and ending with a summary of the previous result section. Arguments should be developed to clarify the ultimate goal, the challenge, the knowledge and technology required to achieve this goal, the statement about this field in general, the significance of this line of research, how this study can serve as an extension of previous work, its implications, and how it can facilitate future research. Thus, it is critical to discuss the review's strengths, weaknesses, and potential clinical applications. The synthesis can be incorporated into this section.
10. I recommend presenting informative figures.
11. Conclusion: The conclusion is well written. Please read my general recommendation: To effectively communicate the manuscript's main message, I recommend dedicating a single paragraph, approximately 150-200 words long, to highlight the author’s extensive and thorough considerations as esteemed experts in their respective fields. This approach would be advantageous because it emphasizes the importance of their efforts to explain the theoretical implications and practical applications of their findings. It is also critical to discuss potential areas for future research, as well as theoretical and methodological aspects that require further development, in order to fully comprehend the significance of this line of research.
12. References: Please follow the guidelines of the journal and cite more references. An article, such as this, typically cites more than 150 references.
The manuscript contains no figures, five tables, and 87 references. I believe that the merit of this study lies in its ability to critically evaluate and synthesize the existing body of research on the relationship between PTSD and cellular aging markers, such as telomere length and telomerase activity. By addressing inconsistencies and methodological challenges across studies, this manuscript provides a nuanced understanding of why these biomarkers have yielded variable results in the context of PTSD. This in-depth analysis not only clarifies current knowledge but also lays the groundwork for more targeted and effective future research, potentially guiding the development of novel therapeutic strategies. I hope that after careful revision, the manuscript meets the journal’s high standards for publication. In addition, I anticipate the author preparing “a detailed point-point rebuttal” to my remarks.
I declare no conflict of interest regarding this manuscript.
Best regards,
Reviewer
References:
- https://onlinelibrary.wiley.com/doi/full/10.1111/j.1471-1842.2009.00848.x
- https://www.edanz.com/excited-science/systematic-scoping-narrative
- https://link.springer.com/chapter/10.1007/978-981-16-5248-6_29
Author Response
1. "First, I would like the author to clarify the type of review article, such as systematic, scoping, or synthetic, and declare it in the title, abstract and the objectives of introduction. Then, I would like the author to make sure there are all elements necessary for a certain type of review paper by using the checklist [1–3]."
Reply: The type of review has been clearly indicated in the title, abstract, objectives (line 393) and review process (line 412). The review followed the standard guidelines for narrative syntheses as indicated in lines 415-416 and references no. 98 and 99.
2. Title: Please provide a concise and informative title that accurately reflects the key message of this study, as this is the most essential aspect of the manuscript. Please consider: "Unraveling the Telomere Puzzle: PTSD and the Complexities of Cellular Aging"; "PTSD and Accelerated Aging: A Critical Reappraisal of Telomere Dynamics".
Reply: The title has been corrected to "Telomere dynamics in post-traumatic stress disorder: a critical synthesis", as per the reviewer's suggestion.
3. Abstract: The abstract is generally well written, but please reconsider the following guidelines: I recommend that the author presents the background, methods, results, and conclusion in a proportional order within 200 words without subheadings. The general background (one to two sentences), specific background (two to three sentences), and current issue addressed in this study (one sentence) should all be included in the background before moving on to the objectives. In this subsection, I would like the author to provide background information, a problem statement, and their reasoning for breaking off. The results section ends with a phrase that places this subsection in a broader context. The conclusion should start with a single sentence that summarizes the main message, such as "Here we show." The author should describe the potential and progress of this study in the field in the first sentence of the conclusion, followed by two to three sentences that provide a broader perspective that any scientist can comprehend. Please refrain from presenting statistical details in the abstract.
Reply: The abstract has been rewritten as per the above suggestions (lines 9-27). No statistical details are given in the abstract.
4. Keywords: Please include ten keywords from Medical Subject Headings (MeSH) in the title and the first two sentences of the abstract.
Reply: Keywords as per MeSH have been included with the revised manuscript (lines 29-30).
5. I highly recommend presenting an informative graphical or video abstract.
Reply: I thank the reviewer for this suggestion. However, given the complex nature of the results being analyzed, including several conflicting findings, this could not be done. Instead, the key findings and results are presented in tables as suggested by the reviewer.
6. Introduction: This section is well written, but please consider expanding this section according to the following guidelines: I recommend that the author presents this section before presenting the details of the following sections, incorporating several paragraphs totaling approximately 1000 words, to provide a comprehensive introduction to the main constructs of this study. This introduction should be understandable to readers from all disciplines and serve to establish the purpose of the current research as well as the specific intent of this study. The author is advised to begin with a general background, then move on to a more specific background, and the current issue addressed in this study, leading to the objectives. The manuscript's objectives should be reached through a coherent and logical progression.
Reply: The introduction has been rewritten, expanded, and organized into four subheadings as per the reviewer's suggestion (lines 32-383).
7. Methods: Please present the method section if this manuscript is systematic, scoping, or synthetic review. This section should start with a brief introduction and then cites additional references to ensure the reliability and integrity of the evidence in the author’s study design and methodology. This section's key elements include a PRISMA flow chart.
Reply: As per the reviewer's suggestion, the methods section has been rewritten with appropriate references (lines 387-444). As PRISMA does not recommend a flow chart for narrative / synthetic reviews, this could not be included.
8. Results: Please present the result section if this manuscript is intended to be systematic, scoping, or synthetic review. I recommend including a paragraph that provides a summary of the findings as the section's conclusion. This section's key elements include a risk of bias assessment.
Reply: The Results section has been reorganized and formatted as per the reviewer's suggestions. Key findings have been included in 1-2 paragraphs for each section and full details are given in Tables 1-5. Study quality (including risk of bias) has been assessed for all studies using the BIOCROSS instrument.
9. Discussion: I would like the author to present the paragraphs of the discussion section totaling around 1,500 words, beginning with an introductory paragraph and ending with a summary of the previous result section. Arguments should be developed to clarify the ultimate goal, the challenge, the knowledge and technology required to achieve this goal, the statement about this field in general, the significance of this line of research, how this study can serve as an extension of previous work, its implications, and how it can facilitate future research. Thus, it is critical to discuss the review's strengths, weaknesses, and potential clinical applications. The synthesis can be incorporated into this section.
Reply: The discussion section has been completely rewritten as per the reviewer's suggestions, including a discussion of limitations and strengths (lines 716-976).
10. I recommend presenting informative figures.
Reply: I thank the reviewer for this suggestion. However, given the complex nature of the results being analyzed, including several conflicting findings, this could not be done. Instead, the key findings and results are presented in tables as suggested by the reviewer.
11. Conclusion: The conclusion is well written. Please read my general recommendation: To effectively communicate the manuscript's main message, I recommend dedicating a single paragraph, approximately 150-200 words long, to highlight the author’s extensive and thorough considerations as esteemed experts in their respective fields. This approach would be advantageous because it emphasizes the importance of their efforts to explain the theoretical implications and practical applications of their findings. It is also critical to discuss potential areas for future research, as well as theoretical and methodological aspects that require further development, in order to fully comprehend the significance of this line of research.
Reply: The Conclusion has been rewritten as suggested by the reviewer in lines 977-991.
12. References: Please follow the guidelines of the journal and cite more references. An article, such as this, typically cites more than 150 references.
The manuscript contains no figures, five tables, and 87 references. I believe that the merit of this study lies in its ability to critically evaluate and synthesize the existing body of research on the relationship between PTSD and cellular aging markers, such as telomere length and telomerase activity. By addressing inconsistencies and methodological challenges across studies, this manuscript provides a nuanced understanding of why these biomarkers have yielded variable results in the context of PTSD. This in-depth analysis not only clarifies current knowledge but also lays the groundwork for more targeted and effective future research, potentially guiding the development of novel therapeutic strategies. I hope that after careful revision, the manuscript meets the journal’s high standards for publication. In addition, I anticipate the author preparing “a detailed point-point rebuttal” to my remarks
Response: As recommended by the reviewer, the revised manuscript now contains 164 relevant references.
Reviewer 2 Report
Comments and Suggestions for Authors
The author provides an update on the relationship between PTSD and TL. The author made an effort to search and organize the literature chronologically. The manuscript was well-written, categorized, and discussed the results by the group—positive, negative, and no association with TL.
I have some concerns with all previous studies of PTSD and TL in that the justification for the study of PTSD and TL. Both PTSD and TL are meaningful clinical outcomes with highly heterogeneous etiology.
PTSD is caused or triggered by different acute and chronic traumatic events such as childhood trauma, rape, combated trauma, and earthquake, but the long-term outcome may be diverse; some may become more resilience after trauma events in the long run, as shown in a longitudinal study by Boks et al. (2014) and multiple studies showing a positive correlation between TL and PTSD symptoms (Table 2). In general, the outcome of trauma is determined by genetics, brain development, early childhood experience (both positive and negative), type, and times of traumatic events. In contrast, while being reduced by exposure to chronic stressors, TL is a highly heritable trait (heritability >0.8). The telomere length is regulated by two opposing mechanisms: attrition and elongation, modulated by telomerase activity, meditation, and dietary behaviors.
Etiologically, I couldn’t see a solid justification for investigating PTSD and TL. In particular, most of the reviewed studies were conducted with a case-control design, which is notoriously known for its confounding effect. As expected by this review, the results were inconsistent.
I have the following comments that may help improve the manuscript.
l I suggest the author or group them by study design, which is important for assessing the results.
l In this type of review, it would be good to use Tables to present data rather than text. For example, in Table 3 and other tables, it would be good to present study information on sample size, study design, covariates, and results. This would allow the audience to assess the data by themselves.
l In section 3.3, what did the author mean by translational research and animal studies? I would call it animal studies; in fact, it was called animal research in 4.1.3
l In 3.2.2., telomerase activity is through to affect the telomere length. Why did the author add literature about telomerase activity and genetic evidence of PTSD without mentioning TL? They seemed irrelevant here.
l Table 4. Animal studies showed something interesting and are expected in that TL is mainly affected by chronic stress rather than acute traumatic stress that induces PTSD.
l Line 322. “Studies that do not control for possible confounders may overestimate the association between PTSD and changes in TL”. The confounding effect could be “overestimate or underestimate.”

Author Response
1. I suggest the author or group them by study design, which is important for assessing the results.
Response: As suggested by the reviewer, the studies have been grouped by design as well as by chronology. Details of study design have been added to the tables.
2. In this type of review, it would be good to use Tables to present data rather than text. For example, in Table 3 and other tables, it would be good to present study information on sample size, study design, covariates, and results. This would allow the audience to assess the data by themselves.
Response: I agree with this important suggestion. The requested details have been added to the tables on human studies and the amount of text in the results section has been minimized.
3. In section 3.3, what did the author mean by translational research and animal studies? I would call it animal studies; in fact, it was called animal research in 4.1.3
Response: I apologize for this error. It has been corrected to "Animal studies" / "Animal research" throughout.
4. Telomerase activity is through to affect the telomere length. Why did the author add literature about telomerase activity and genetic evidence of PTSD without mentioning TL? They seemed irrelevant here.
Response: I apologize for this error which was due to a mistake on my part in copying and pasting text from the draft to the final manuscript. The sections on TL and telomerase have been reformatted and reorganized.
5. Animal studies showed something interesting and are expected in that TL is mainly affected by chronic stress rather than acute traumatic stress that induces PTSD.
Response: I thank the reviewer for this important insight. This has been discussed in the Discussion section (lines 845-858).
6. Line 322. “Studies that do not control for possible confounders may overestimate the association between PTSD and changes in TL”. The confounding effect could be “overestimate or underestimate.”
Response: This has been corrected as suggested.
Round 2
Reviewer 1 Report
Comments and Suggestions for Authors
4 December 2024
The 2nd review report on the manuscript, titled ‘Telomere length and telomerase in post-traumatic stress disorder: an update and a reappraisal of the evidence’ by Rajkumar, RP submitted to Biomedicines
Manuscript ID: biomedicines-3193437
Dear Author,
I am pleased that the authors have addressed the issues raised in the previous round. Currently, the manuscript is a well-written research paper with informative layouts, which reviews the relationship between post-traumatic stress disorder and cellular aging, focusing on telomere length and telomerase activity as potential biomarkers of accelerated aging. I believe the manuscript meets the journal’s high standards for publication. I am looking forward to seeing more papers written by the same authors.
Thank you!
I declare no conflict of interest regarding this manuscript.
Best regards,
Reviewer
Author Response
Thank you for your valuable suggestions in improving the quality of this manuscript.
Reviewer 2 Report
Comments and Suggestions for Authors
This manuscript was drastically revised, and tracking changes make it difficult to read. I suggest the authors make a clean version and resubmit it as a new manuscript.
1. The instruction needs to be shortened.
2. The method may require a diagram to demonstrate the search keywords, and how the article was excluded step by step.
3. In each section of the results, there were so many descriptives; it would be very helpful if the authors provided a summary paragraph for that section.

Author Response
1. I suggest the authors make a clean version and resubmit it as a new manuscript.
Response: I thank the reviewer for this suggestion. A clean version of the manuscript has been resubmitted, with only the changes requested by the current reviewer marked using the tracking function.
2. The instruction needs to be shortened.
Response: I thank the reviewer for their suggestion. The introduction was lengthened at the request of Reviewer 1, whose report can be consulted for clarification. However, as requested, the introduction has been further revised and shortened by around 700-800 words; the changes are marked in the revised manuscript.
3. The method may require a diagram to demonstrate the search keywords, and how the article was excluded step by step.
Response: I thank the reviewer for this important suggestion. A diagram (Figure 1) has been added to clarify this aspect of the review.
4. In each section of the results, there were so many descriptives; it would be very helpful if the authors provided a summary paragraph for that section.
Response: I agree with the reviewer's suggestion. Summary paragraphs have been added as requested at lines 436-439, lines 491-498, and lines 562-564 in the revised manuscript.